# Learning Simultaneous Navigation and Construction in Grid Worlds

**Wenyu Han, Haoran Wu, Eisuke Hirota, Alexander Gao, Lerrel Pinto, Ludovic Righetti, Chen Feng***
New York University

## Abstract

We propose to study a new learning task, *mobile construction*, to enable an agent to build designed structures in 1/2/3D grid worlds while navigating in the same evolving environments. Unlike existing robot learning tasks such as visual navigation and object manipulation, this task is challenging because of the interdependence between accurate localization and strategic construction planning. In pursuit of generic and adaptive solutions to this partially observable Markov decision process (POMDP) based on deep reinforcement learning (RL), we design a Deep Recurrent Q-Network (DRQN) with explicit recurrent position estimation in this dynamic grid world. Our extensive experiments show that pre-training this position estimation module before Q-learning can significantly improve the construction performance measured by the intersection-over-union score, achieving the best results in our benchmark of various baselines including model-free and model-based RL, a handcrafted SLAM-based policy, and human players. Our code is available at: https://ai4ce.github.io/SNAC/.

## 1 Introduction

Intelligent agents, from animal architects (e.g., mound-building termites and burrowing rodents) to human beings, can simultaneously build structures while navigating inside such a dynamically evolving environment, revealing robust and coordinated spatial skills like localization, mapping, and planning. Can we create artificial intelligence (AI) to perform similar *mobile construction* tasks?

To handcraft such an AI using existing robotics techniques is difficult and non-trivial. A fundamental challenge is *the tight interdependence of robot localization and long-term planning for environment modification*. If GPS and techniques alike are not available (often due to occlusions), robots have to rely on simultaneous localization and mapping (SLAM) or structure from motion (SfM) for pose estimation. But mobile construction violates the basic *static-environment* assumption in classic visual SLAM methods, and even challenges SfM methods designed for *dynamic scenes* (Saputra et al., 2018). Thus, we need to tackle the interdependence challenge to strategically modify the environment while efficiently updating a memory of the evolving structure in order to perform accurate localization and construction, as shown in Figure 1.

Deep reinforcement learning (DRL) offers another possibility, especially given its recent success in game playing and robot control. Can deep networks learn a generic and adaptive policy that controls the AI to build calculated structures as temporary localization landmarks which eventually evolve into the designed one? To answer this question, we design an efficient simulation environment with a series of mobile construction tasks in 1/2/3D grid worlds. This reasonably simplifies the environment dynamics and sensing models while keeping the tasks nontrivial, and allows us to focus on the aforementioned interdependence challenge before advancing to other real-world complexities.

To show the tasks in grid worlds are still non-trivial and challenging, we benchmark the performance of several baselines, including human players, a handcrafted policy with rudimentary SLAM and planning, some model-free DRL algorithms which have achieved state-of-the-art performance in other learning tasks (see Table 1 for comparisons), and a model-based DRL using Deep Q-Networks (DQN) augmented with Monte Carlo tree search (MCTS) as planning mechanism. Although our

---

*Corresponding author, cfeng@nyu.edu

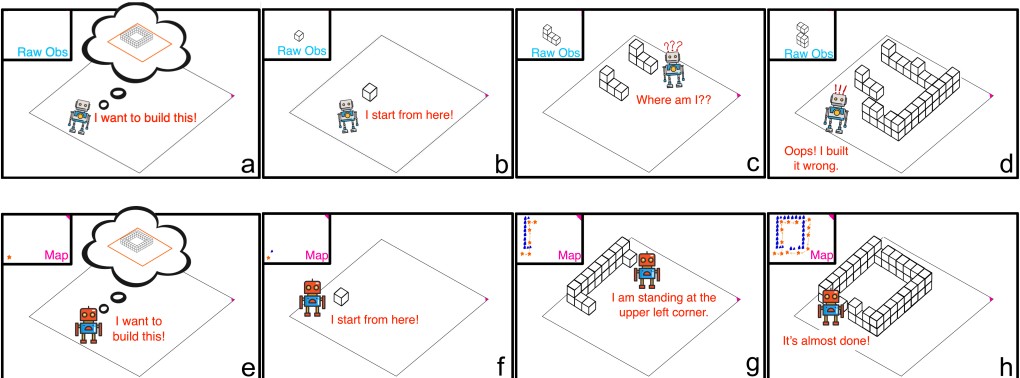

Figure 1: Challenge in mobile construction. An AI needs to navigate in an environment (square area) and build a structure according to a design (stacked cubes in a/e). The AI in (a-d) builds poorly because it learns localization and construction jointly from raw observations. Specifically, similar structures seen in (c) confuse the AI due to the wrong localization. In contrast, a better construction is achieved in (e-h) using a pre-trained localization even in this non-static environment.

| | Loc | Plan | Env-Mod | Env-Eval |
|---|---|---|---|---|
| Robot Manipulation (Fan et al., 2018; Yang et al., 2019; Labbé et al., 2020; Li et al., 2020) | ✗ | ✓ | ✓ | ✗ |
| Robot Locomotion (Duan et al., 2016) | ✗ | ✗ | ✗ | ✗ |
| Visual Navigation (Zhu et al., 2017; Gupta et al., 2017; Mo et al., 2018; Zeng et al., 2020) | ✓ | ✓ | ✗ | ✗ |
| Atari (Mnih et al., 2013) | ✗ | ✓/✗ | ✓ | ✗ |
| Minecraft (Oh et al., 2016; Guss et al., 2019; Platanios et al., 2020) | ✗ | ✓/✗ | ✓ | ✗ |
| First-Person-Shooting (Lample & Chaplot, 2017) | ✓ | ✗ | ✗ | ✗ |
| Real-Time Strategy Games (Synnaeve et al., 2016; Jaderberg et al., 2019) | ✓ | ✓ | ✓ | ✗ |
| Physical Reasoning Bapst et al. (2019); Bakhtin et al. (2019) | ✗ | ✓ | ✓ | ✗ |
| **Mobile Construction (Ours)** | ✓ | ✓ | ✓ | ✓ |

Table 1: Mobile construction vs. existing learning tasks. **Loc**: robot localization. **Plan**: long-term planning. **Env-Mod**: environment structure modification. **Env-Eval**: evaluation of the accuracy of environment modifications. This shows the novelty of the task with fundamentally different features than typically benchmarked tasks, requiring joint efforts of localization, planning, and learning.

tasks may seem similar to other grid/pixelized tasks such as Atari games (Mnih et al., 2013), the results reveal the significantly worse performance of those baseline algorithms, especially model-free DRL methods, than the human baseline.

A recent study (Stooke et al., 2021) found that decoupling representation learning from RL policy learning can outperform the joint learning of the two in standard RL algorithms. Inspired by this, we propose to pre-train an explicit position estimation module using recurrent neural networks in the above DRL baselines. Our experiment results show that this proposed method outperforms other RL baselines.

In summary, our contributions include:

- a suite of novel and easily extensible learning tasks focusing on the interdependent localization and planning problem, which are released as open-source fast Gym environments;
- a comprehensive benchmark of baseline methods, which demonstrates the learning challenge in these tasks;
- an effective approach which combines DRQN with an explicit position estimation deep network, outperforming other baselines;
- a detailed ablation study providing insights about the causes of the challenge, which could inspire future algorithms to solve the problem more effectively.

## 2 RELATED WORKS

**RL baselines**. With the great success of model-free RL methods in game-playing (Mnih et al., 2013; Silver et al., 2016) and robot control (Cheng et al., 2019; Zhang et al., 2015), we consider a family

of common model-free RL methods as baselines. Since our tasks are in grid worlds, similar to many Atari games such as the Pac-Man, our first choice is DQN (Mnih et al., 2013). However, DQN has limited capability to tackle POMDP due to its poor latent state representation from long-term history, we benchmark DRQN (Hausknecht & Stone, 2015) which uses recurrent Q-network for integrating historical information. Since our tasks may require explicit long-term planning, similar to (Silver et al., 2016; Bapst et al., 2019), we also include a model-based method which uses DQN as the prior and MCTS (Coulom, 2006) as the planning mechanism.

In addition, we add an actor-critic-based baseline, Proximal Policy Optimization (PPO) (Schulman et al., 2017). Whereas the standard policy gradient method performs one gradient update per data sample, PPO enables multiple epochs of minibatch updates by a novel objective with clipped probability ratios. We also include Rainbow (Hessel et al., 2017), SAC (Haarnoja et al., 2018), and DRQN combined with Hindsight Experience Replay (HER) (Andrychowicz et al., 2017) as additional baselines in supplementary Section B.

**Representation learning for RL**. Although the aforementioned RL methods work well on many robot learning tasks, it is still challenging to end-to-end learn meaningful representations and control policies jointly via DRL (Stooke et al., 2021). Recently, a variety of studies improve the performance of RL methods through representation learning, such as internal feature learning (Lample & Chaplot, 2017), and auxiliary tasks (Jaderberg et al., 2016; Mirowski et al., 2016). Echoing these studies, we also find in our experiments that direct end-to-end DRL from raw observations without any representation learning could not solve our task efficiently. Therefore, we propose to introduce a position-related representation learning module in our DRL framework, and pre-train this module to decouple its training from DRL.

Position-related representation learning is not new in DRL. For example, DRL-based visual navigation studies (Zhu et al., 2017; Chaplot et al., 2020; Datta et al., 2021) use neural networks to extract from visual inputs latent representations encoding position information for RL agents to navigate in a static environment. Differently, our task involves a dynamically evolving environment. While work like (Hoeller et al., 2021) indeed navigates robots in dynamic environments using latent state representations in RL, distinctively, our task further involves active modification of the environment. Thus, we do not include this line of work as our baseline.

**Non-learning baseline**. Besides the above methods, one may wish to see the performance of more classical approaches such as either dynamic (Saputra et al., 2018) or active (Mu et al., 2016) visual SLAM which has been studied for decades in robotics, even though they are not designed to work in dynamic environments like ours, nor do they study the planning for better construction accuracy. Therefore, we implement a naive handcrafted policy (pseudo code in Algorithm 1 in the supplementary Section B) with basic localization and planning modules as a non-learning baseline. The localization module borrows the idea from visual SLAM (Saputra et al., 2018) which relies on finding the common features through successive images to estimate the robot's pose. The planning module simply controls the robot to always build at the nearest possible location.

## 3 MOBILE CONSTRUCTION IN GRID WORLD

### 3.1 TASK OVERVIEW

We formulate a mobile construction task as a 6-tuple POMDP $\langle \mathcal{S}, \mathcal{A}, \mathcal{T}, \mathcal{O}, \mathcal{R}, D \rangle$, in which a robot is required to accurately create geometric shapes according to a design $D$ in a grid world. The state space $\mathcal{S}$ is represented as $\mathcal{S} = \mathcal{G} \times \mathcal{P}$, where $\mathcal{G}$ is a space of all possible grid states $G$ storing the number of bricks at each grid, and $\mathcal{P}$ is a space of all possible robot positions $p$ in the grid world.

At each time step, the robot takes an action $a \in \mathcal{A}$, either moving around or dropping a brick at or near its location. Moving a robot will change its location according to the *unknown* probabilistic transition model $\mathcal{T}(p'|p, G, a)$. Dropping a brick will change the grid state $G$ at (for 1D & 2D tasks) or near (for 3D tasks) position $p$ *without any uncertainty* for simplicity. This is reasonable because, in mobile construction settings, motion uncertainty is often the key challenge to ensure accuracy (Sandy et al., 2016).

The robot can make a local observation $o \in \mathcal{O}$ of $G$ centering around its current location, with a sensing region defined by a half window size $W_s$. We pad the constant value $-1$ outside the grid world boundary to ensure valid observations in $\mathcal{P}$, which is distinguished from empty ($= 0$) or filled

($> 0$) grid states. This could help the robot localize itself near the boundary. Finally, the design $D \in \mathcal{G}$ is simply a goal state of the grid world the robot needs to achieve, and $\mathcal{R}(s, a; D)$ is the reward function depending on this design.

The aforementioned interdependence challenge in localization and planning is reflected via two factors in this setting. First, partial observability makes robot localization necessary. Second, the environment uncertainty in $\mathcal{T}$ simulates real-world scenarios where the motion control of the mobile robot is imperfect and the odometry is error-prone. We implement this by sampling the robot's moving distance $d$ in each simulation time step from a uniform distribution.

## 3.2 Task Variations

Next, we specialize the above formulation into a suite of progressively more challenging tasks which varies in the setup of dimension, design $D$ type, and obstacle mechanism, as summarized in Table 2 and Figure 2.

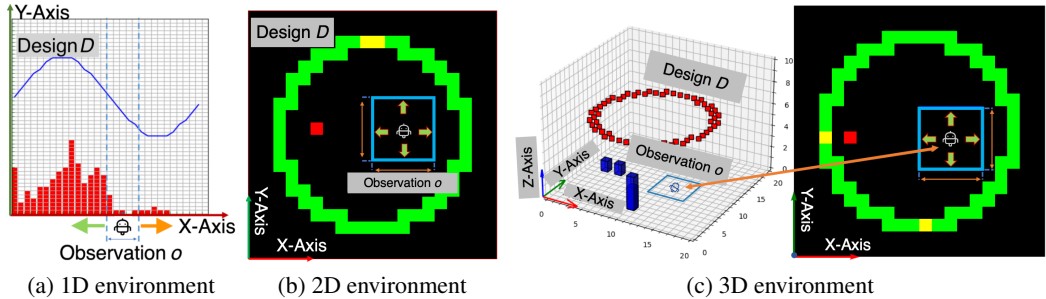

(a) 1D environment      (b) 2D environment      (c) 3D environment

Figure 2: **(a)** 1D grid world environment: a robot moves along the x-axis and drops bricks (red) along -y direction. The two vertical blue dash lines indicate its sensing region for vector $o$. The $o = (5, 1, 1, 0, 1)$ in this case. The blue curve is the design $D$. **(b)** 2D environment: a robot moves in a 2D plane and lays bricks (red/yellow: in/correct dropping). The blue box is its sensing region for $o$, which is a 2D array with size $(2W_s + 1)^2$. The green ring is the ground-truth design $D$. **(c)** 3D environment: a robot moves in a 2D x-y plane and lays bricks (dark blue cubes). The bottom left/right is the 3D/top view. The blue box is similar to 2D. The red ring in the bottom left is the topmost surface of $D$.

| Dimension | 1D | 2D | 3D |
|---|---|---|---|
| **Action** $a$ | Move-left/-right, drop-brick at current location | Move-left/-right/-up/-down, drop-brick at current location | Move-left/-right/-forward/-backward, drop-brick on left/right/front/rear side |
| **Grid state** $G$ | $\mathbb{R}^W$ | $[0, 1]^{W \times H}$ | $\mathbb{R}^{W \times H}$ |
| **Observation** $o$ | $\mathbb{R}^{2W_s + 1}$ | $[0, 1]^{(2W_s+1) \times (2W_s+1)}$ | $\mathbb{R}^{(2W_s+1) \times (2W_s+1)}$ |
| **Constant & Variable** | ✓ | ✓ | ✓ |
| **Dense & Sparse** | ✗ | ✓ | ✓ |
| **Obstacle** | ✗ | ✗ | ✓ |

Table 2: Detail setups of each character for 1/2/3D environment. Each character is described in Section 3 and illustrated in Figure 2. In 1D environments, the grid state $G \in \mathbb{R}^W$ is a vector, where $W$ is the width of the environment and $o \in \mathbb{R}^{2W_s+1}$ is a vector with size $2W_s + 1$. The grid state $G$ for 2D and 3D environment are 2D binary matrix and 2D matrix with width $W$ and height $H$ respectively. We only add obstacle mechanism in 3D environment.

**Dimension.** We vary our tasks by the grid world dimension in 1/2/3D respectively. The robot is restricted to moving along the x-axis in 1D and x-y plane for 2D and 3D environments.

**Constant** vs. **Variable Design**. The design of our tasks could be either constant or variable. The constant design task requests a robot to build the same shape $D$ in all episodes in both training and testing. For the variable design task, the design $D$ will vary for each episode. For a variable design task, to account for real-world situations where robots should have access to the design, we augment the local observation with additional information such as the design $D$ as a 4-tuple $o_{env} = \langle o, N_s, N_b, D \rangle$, where $N_s$ is the number of actions the robot has taken, and $N_b$ is the number of bricks used.

**Dense** vs. **Sparse Design.** A dense design is a solid shape, and a sparse design is an unfilled shape (Figure A.2e and Figure A.2f).

**Obstacles.** Different from 1D and 2D tasks where the robot size is ignored, the robot in the 3D tasks occupies 1 grid so its motion could be obstructed by the built bricks.

**Stop criteria.** Each episode ends when $N_s = N_{smax}$ or $N_b = N_{bmax}$, where the former is the maximum number of steps and the latter the maximum number of bricks (the integrated area of the design $D$ within each episode). We set $N_{smax}$ reasonably large to ensure the design completion. In addition to the same stop criteria as in the 1D/2D cases, the game will stop when the robot is obstructed by the built bricks and cannot move anymore in a 3D environment.

## 4    RECURRENT POSITION ESTIMATION IN DYNAMIC ENVIRONMENTS

After formulating mobile construction tasks, we benchmarked our naive model-free DRL baselines on these tasks. We found that these DRL policies perform worse than our human baselines, especially on 2D and 3D tasks. We believe the low performance is due to the difficulty in learning meaningful representations and an effective control policy jointly via RL training alone. Especially, the aforementioned interdependence challenge of mobile construction tasks requires the agent to localize itself in a dynamic environment where the surrounding structure could change after the agent build a new brick. Inspired by some recent studies (Stooke et al., 2021; Lample & Chaplot, 2017; Jaderberg et al., 2016; Mirowski et al., 2016) which decouple representation learning from RL, we propose our method which combines (1) a pre-trained localization network (L-Net) to estimate the current agent position and (2) a DRQN to select the best action based on the predicted positions and observations. Section 4.1 introduces our proposed method and section 4.2 explains the training process of L-Net.

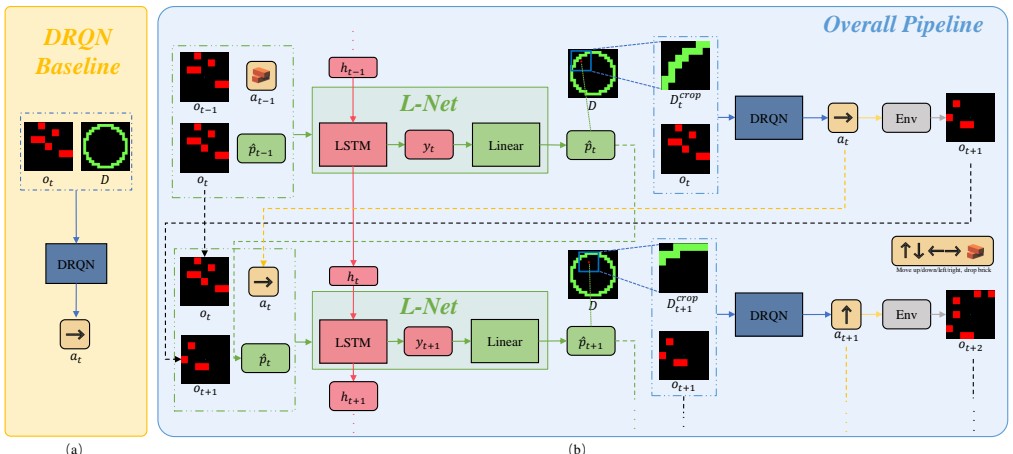

Figure 3: Illustration of the DRQN baseline and our proposed method. **(a)** We use the standard DRQN model which takes the raw observation $o_t$ and goal design $D$ as input and outputs an action $a_t$ in each time step $t$ as our baseline. **(b)** Our pipeline contains two main components: (1) a pre-trained L-Net which is an LSTM taking the $o_{t-1}, a_{t-1}$, hidden state $h_{t-1}$, and the predicted robot position $\hat{p}_{t-1}$ from the previous frame and $o_t$ of the current frame as input and outputting a predicted $\hat{p}_t$ and hidden state $h_t$ in the current step. Then, we crop a local design $D_t^{crop}$ centering at the predicted position $\hat{p}_t$ from goal $D$. This $D_t^{crop}$ has the same size as $o_t$. (2) we feed the $D_t^{crop}$ and $o_t$ to a DRQN to make an action $a_t$ of current step. The detailed training process of L-Net is explained in section 4.2, and shown in supplementary Figure B.1

### 4.1    DRQN WITH L-NET

Our method is shown in Figure 3, consisting of two main components: (1) an L-Net which is pre-trained via supervised learning, using an LSTM to predict position $\hat{p}_t$ at each time step $t$; (2) a standard DRQN which takes predicted positions $\hat{p}_t$, observations $o_t$, and cropped goal design $D^{crop}$ as input and outputs the actions $a_t$ at current frame.

Let us use the 2D dynamic task as an example to explain the overall pipeline. Firstly, the raw observations $o_{t-1}, o_t$ of the previous and current time steps respectively, are flattened to vectors. Then these two vectors are concatenated with the predicted position $\hat{p}_{t-1}$ and the action $a_{t-1}$ in the previous time step. This concatenated vector $z$ and hidden state $h_{t-1}$ are fed to an LSTM to output a hidden feature $y_t$ and its hidden state $h_t$. Finally, a linear layer is used to map the input hidden feature $y_t$ to a predicted position $\hat{p}_t$. The L-Net which is composed of the LSTM and the linear layer is pre-trained through supervised learning.

As shown in Figure 3a, the observation $o_t$ and the design $D$ are given to the DRQN to output an action $a_t$ in the baseline. We tried this way of inputting information to DRQN similarly in our pipeline in Figure 3b, and we additionally input the predicted position $p_t$ to the DRQN in our method at the beginning. But the resulting performance is not significantly better than our model-free RL baselines. This suggests that the raw input (the predicted robot position and the complete design) is not an efficient representation for DRQN-based training.

Inspired by our human volunteers who choose to only refer to a local region of the $D$ near the current position to make decisions when solving our tasks, we crop a local design $D_t^{crop}$ centering at the predicted position $\hat{p}_t$ from the complete design $D$. Then, the $D_t^{crop}$, $\hat{p}_t$, and $o_t$ are given to the DRQN-based policy network to output an action. Note that when training this pipeline, the pre-trained L-Net is frozen and only the initial position $p_0$ is assumed known. All other positions $\hat{p}_t|_{t=1,2,...}$, used in each DRQN step are predicted by L-Net based on predicted positions $\hat{p}_{t-1}$ from its previous time step.

## 4.2    L-NET PRE-TRAINING

Since the L-Net is trained apart from the overall pipeline training by supervised learning, we collect a dataset which contains 30000 episodes of games. Each game leads to a replay buffer $\{M_i\}|_{i=1,2,...,N}$, where $M_i = (o_{i-1}, a_{i-1}, p_{i-1}, o_i)$. Each data in such a buffer is collected by letting the agent perform random actions until an episode ends. When training the L-Net, we randomly sample a chunk of data $\{M_i\}|_{i=t+1,t+2,...,t+L}$, where $L$ is a constant sequence length, from the entire sequence. As shown in supplementary Figure B.1, we firstly feed the data $M_{t+1}$ and a randomly initialized hidden state $h_t$ to the L-Net that predicts position $\hat{p}_{t+1}$ and its hidden state $h_{t+1}$. Note that we always use the first ground-truth (GT) position $p_t$ as input and the input positions of following steps are all outputs from previous steps. This means we *do not* perform teacher-forcing to train the LSTM. Then, we just repeat $L$ steps and compute $L$ predicted positions $\hat{p}_i|_{i=t+1,t+2,...,t+L}$. Finally, the L-Net is trained by minimizing the L2 loss between the predicted positions and GT positions.

## 5    EXPERIMENTS

We conduct comprehensive experiments to test baselines and our proposed method on mobile construction tasks. In this section, we only pick representative baselines from different classes to show the results. All experiment results are listed in supplementary Table 7 and the best qualitative results are shown in supplementary Figure C.3. The evaluation metric, reward function, and training protocol are described in this section.

**Baselines.** We choose two Q-learning methods, an actor-critic method, a search-based method, a non-learning method, and human players as representative baselines. Detailed architecture designs and hyperparameter setups for each algorithm are explained in the supplementary Section B.

- **DQN**. For constant design tasks, we use an MLP-based Q network. For variable design tasks, the design $D$ is mapped to a feature vector by a 3 convolutional layers.
- **DRQN**. We add one recurrent LSTM layer to the Q network as the DRQN baseline.
- **PPO**. We benchmark discrete PPO in the Stable Baselines (Hill et al., 2018).
- **DQN+MCTS**. For this model-based method, we simply combine DQN with standard MCTS algorithm, similar to Bapst et al. (2019).
- **Handcrafted policy.** It consists of basic localization and planning modules.
- **Human**. A simple GUI game that allows a human player to attempt our tasks in identical environments with the same limitations of partial observability and step size uncertainty. They act as the agent, using the ARROW keys to maneuver, and SPACE to drop a brick.

**Evaluation metric:** we use the Intersection over Union (IoU) score as our evaluation criteria which is measured between the terminal grid state $G^T$ and the ground-truth design $D$. Here, we use IoU as our evaluation criteria because it is more straightforward for us to evaluate the quality of the built structure to the ground-truth design $D$. The IoU is defined as:

$$IoU = \frac{G^T \cap D}{G^T \cup D} = \sum_{p \in \mathcal{P}} \frac{\min(G^T(p), D(p))}{\max(G^T(p), D(p))}. \tag{1}$$

**Reward function.** The reward function $\mathcal{R}$ for 1D and 3D environments is shown in Table 3. For the 2D environment, $\mathcal{R}$ is simply designed as: the agent gets 5/0 for dropping bricks at correct/wrong positions and gets 0 for movement. Note that in the 2D environment we tried to add a similar penalty as in Table 3 for incorrect brick dropping, but we found this led to more frequent actions of moving instead of dropping bricks because of the higher chance of negative rewards of the latter (especially for sparse plans). Consequently, this led to bad performance of all tested methods. Therefore, we removed the penalty from the reward function in all 2D environments.

**Training protocol.** To validate the proposed framework and its robustness, all baselines are trained with the same set of 4 random seeds and averaged results are reported. For the constant design tasks in 1D/2D/3D, we test the trained agent

|  | Drops brick on $D$ | Drops brick below $D$ | Drops brick over $D$ | Move | Obstructed |
|---|---|---|---|---|---|
| 1D | 10 | 1 | -1 | 0 | ✗ |
| 3D | 10 | 1 | -1 | 0 | -100 |

Table 3: Reward function for 1D and 3D tasks.

for 500 times for each task and report the average and the standard deviation (stddev) of IoU scores among 500 tests. For the variable design tasks, we randomly generated 500 ground-truth designs and split them to 8/1/1 for training/validation/testing. Following the same protocol as constant design tasks, we test all baselines on the testing set for 500 times for each task and report the average and the stddev of IoU scores among 500 tests.

## 5.1 BENCHMARK RESULTS

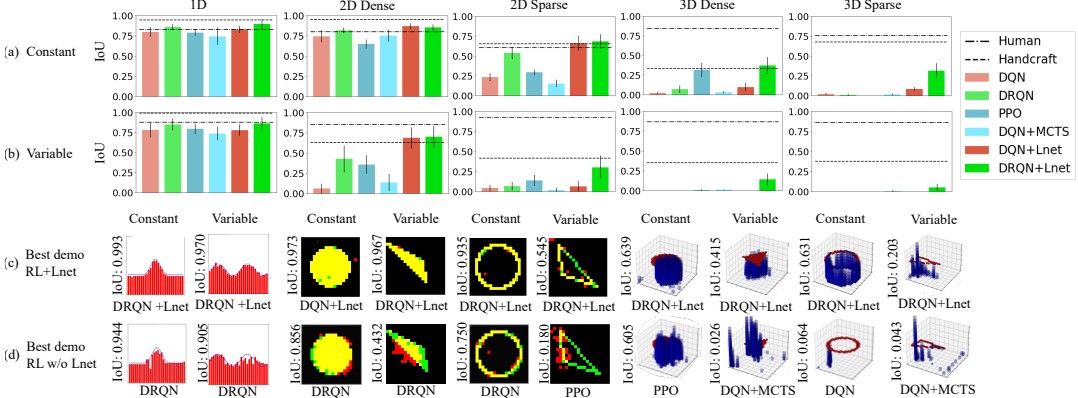

Figure 4: Experiment results of baselines and our method. (a) & (b) present the average and the stddev (errorbars) of IoU of examined methods on constant and variable design tasks in 1D/2D/3D environments. (c) & (d) show the *best* qualitative results of RL w/ and w/o L-Net, respectively.

Figure 4 summarizes the quantitative testing results and the *best* qualitative testing results of the baselines and our method. In general, we see that the performance of each method drops as the dimension of the environment increases. This clearly shows that the larger the exploration space is and that the more complex the construction mechanisms are (e.g., obstacles in 3D tasks, further studied in Section 5.2), the worse the tested method performs. Especially in 3D environments, most of the baselines learn ineffective policies or fail to learn any policies. Meanwhile, most of the methods consistently perform worse on the sparse design tasks than on the dense ones. We believe that such poor performance is due to the much sparser reward signals in sparse design tasks.

Moreover, we notice that all RL baselines have more difficulties dealing with variable design tasks compared to constant ones. We posit that the RL baselines lack the capacity to learn effective representations for variable designs. DRQN is expected to be the best among all RL baselines in

most of the tasks because it can learn better from a long period of history. Compared with all examined RL baselines, our proposed method achieves better performance consistently in all tasks. In some cases, it can even surpass the human baseline and the handcrafted policy which relies on prior knowledge of the environment setups. Next, we will compare them in detail.

**RL w/ v.s. w/o L-Net**. Besides the proposed method described in section 4, we also include another baseline which combines DQN with L-Net to further test the effectiveness of our method. We find that RL w/ L-Net methods outperform all other RL baselines consistently among all tasks. For 1D constant tasks, the DRQN w/ L-Net achieves 0.896 ave IoU score, which even surpasses the human baseline (IoU 0.83). It also performs the best among other learning-based baselines in 1D variable tasks. As shown in Figure 4(c), DRQN w/ L-Net can perfectly build the Gaussian-shaped design. In contrast, most of the RL baselines are only able to build a flattened shape (See the first column of Figure C.3 in supplementary). All these suggest the importance of representation learning that enables a reasonable position estimation, even in our dynamic environments.

On the 2D constant dense task, all methods reach similar performance. However, the gap between RL w/ L-Net and RL w/o L-Net becomes significantly larger in more difficult 2D tasks, such as the 2D constant sparse task. The DRQN w/ L-Net can achieve a reasonable performance of median IoU 0.301 and best IoU 0.545 in the 2D variable sparse task where others only learn very limited policies. The qualitative results in Figure 4(c) show that DRQN w/ L-Net can complete half of the triangle shape in best cases while RL baselines can only drop a few bricks around the goal design (see the 6th column of Figure 4(d)). This indicates that with the help of position prediction by the pre-trained L-Net, DRQN is able to learn a better construction policy.

Similarly, the DRQN w/ L-Net method outperforms RL baselines in 3D tasks. From the bar chart of Figure 4(a), we can see that DRQN w/ L-Net can still learn reasonable policies which achieve average IoU 0.3074 and 0.315 on the 3D constant dense and sparse tasks. Our method is almost able to construct the complete hollow cylinder in Figure 4(c). Moreover, it is remarkable that DRQN w/ L-Net manages to learn a rough construction policy while other RL baselines fail to learn any useful policies. We conclude that explicit localization via representation learning is crucial for tackling 3D tasks, though the performance is still limited due to the potential challenges of obstacles.

**Handcrafted policy**. Although our handcrafted method performs seemingly well on 1D tasks, one should not draw a hasty conclusion that it must be the ultimate direction for solving mobile construction. Because the good performance of this method relies on prior knowledge of the distribution parameters of the probabilistic transition model $\mathcal{T}$: our handcrafted localization has a very low error rate (especially in 1D settings) by leveraging this privileged information, which is not accessible in other baseline methods and in reality. Moreover, from Figure 4, we can see that the obstacle mechanism, the sparse designs, and the environment uncertainty in 2D and 3D are still challenging for this method. We believe learning-based methods have a good potential to adaptively address these variations, while solving them one by one via different handcrafted methods is less effective.

**Human baseline**. We collected 30 groups of human test data, totaling 490 episodes played, and report the average IoU shown in the Figure 4 with dot line. All humans were required to play the same maximum number of steps for each game as other baselines. From the feedback, we found that humans could learn effective policies (building landmarks to help localization) very efficiently in at most a few hours, which is much sooner than training an RL model.

## 5.2 ABLATION STUDY

We performed ablation studies to comprehensively analyze the reasons associated with the poor RL baseline performances on 2D and 3D

| Shape | IoU | 2D | | | | | 3D | |
|---|---|---|---|---|---|---|---|---|
| | | **DRQN** | +GPS(↑) | -Uncertainty(↑) | +Obstacle(↓) | +Landmark(↑) | **DRQN** | -Obstacle(↑) |
| Dense | Avg | **0.818** | +0.033 | +0.027 | -0.334 | +0.154 | **0.071** | +0.728 |
| | Stddev | **0.032** | -0.005 | -0.032 | +0.037 | -0.002 | **0.048** | -0.009 |
| | Min | **0.724** | +0.027 | +0.121 | -0.591 | +0.156 | **0** | +0.714 |
| Sparse | Avg | **0.538** | +0.402 | +0.238 | -0.306 | +0.406 | **0.009** | +0.184 |
| | Stddev | **0.078** | -0.037 | -0.078 | -0.010 | -0.043 | **0.015** | +0.037 |
| | Min | **0.2** | +0.186 | +0.576 | -0.163 | +0.56 | **0** | 0 |

Table 4: Ablation study on constant design tasks using DRQN. ↑/↓: expecting better/worse performance.

tasks. We identify four potential challenges: (1) the obstacles in 3D environments, (2) the lack of localization information, (3) the step size $d$ uncertainty, and (4) the lack of landmarks in the environment for localization. We use DRQN results on constant tasks as the basis of this ablation study and all the ablation experiments were conducted using the same setup and test criteria described in Section 5. We also try imitation learning to train DRQN with expert experience and conduct an ablation study for Rainbow (see supplementary Section C).

**Obstacles.** To explore the influence of the obstacles, we conducted two experiments: adding the obstacle mechanism in a 2D task and removing it from the 3D one (Figure 5). As shown in Table 4, the performance of the 3D task increases significantly from 0.071 to 0.799 for dense designs. Similarly, when we add obstacles into 2D, the IoU score drops by more than 40%. These results suggest that the presence of obstacles is an important reason for the poor performance on 3D tasks.

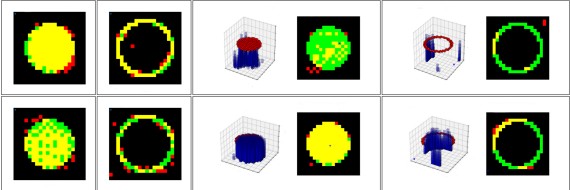

Figure 5: Influences of obstacles. Top rows show the baseline results on 2D and 3D constant tasks. The first two images of bottom row show results of adding obstacle to 2D world. Compared with baseline, the performance drops notably. The last two images of bottom row show results of removing obstacle from 3D world. The performance increases significantly, especially for 3D dense design task.

**Localization.** From the previous experiments, we conclude that good position estimation via representation learning could help RL learn better construction policies. Moreover, it is interesting to test what are RL methods' performance upper bound with the help of perfect localization. To test this, we provided GT position $p$ for DRQN on 2D constant tasks. From Table 4, we can see that with the help of position information, 2D constant sparse is perfectly solved (see Figure 6a).

**Environment uncertainty.** Besides the limited sensing range, the random step size could be another reason for the poor performance of baselines. Therefore, we conduct an experiment where the step size uncertainty is removed. From Table 4, we can see that the IoU increases by more than 40% compared to the sparse designs (see Figure 6b).

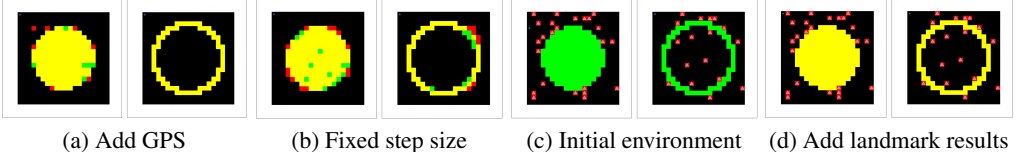

(a) Add GPS      (b) Fixed step size      (c) Initial environment      (d) Add landmark results

Figure 6: Influence of localization and environment uncertainty. Either adding GPS in (a) or removing step size uncertainty in (b) largely improves DRQN on sparse design tasks compared with the first two columns on the top row of Figure 5.Influence of landmarks. (c) We randomly add landmarks in the initial environment (marked as a gray triangle). (d) Compared with the baseline, adding landmarks improves DRQN obviously on both dense and sparse design tasks.

**Landmarks.** An empty initial environment lacks landmarks widely used in SLAM methods for the robot to localize itself. This could be another potential challenge of mobile construction tasks. We conduct an experiment: randomly adding some landmarks in the initial environment. From Table 4, we can see that IoU increases dramatically both on dense and sparse tasks (see Figure 6c and 6d).

# 6 CONCLUSION AND FUTURE WORK

To stimulate the joint effort of robot localization, planning, and deep RL research, we proposed a suite of mobile construction tasks, where we benchmarked the performance of common model-free and model-based deep RL algorithms, a handcrafted policy with basic localization and planning, and human baselines. Meanwhile, we propose our method which incorporates a pre-trained L-Net and DRQN to solve these tasks. The experiment results indicate that learning an explicit position estimation can effectively improve the performance of RL methods on mobile construction tasks. Although the performance of the proposed method is limited on the hardest tasks, we believe augmenting RL frameworks with representation learning is a promising direction to solve the interdependence challenge in mobile construction tasks. We believe that the limitations of current methods could be overcome by designing a localization function which is able to localize more precisely in dynamic environments. In the future, we plan to further extend our mobile construction task suite with more features such as allowing a new action of explicitly placing landmarks, physics-based simulation in continuous worlds, and multi-agent mobile construction.

## 7 REPRODUCIBILITY STATEMENT

To make all our experiment results reproducible, we submit codes with hyperparameters used for each method on each task in the supplementary material. We also provide an instruction on how to use our code to reproduce the experiment results.

## 8 ACKNOWLEDGEMENT

The research is supported by NSF CPS program under CMMI-1932187. The authors gratefully thank our human test participants, Dongdong Liu and Armand Jordana for implementing the SAC and PPO baselines and the helpful comments from Bolei Zhou, Zhen Liu, and the anonymous reviewers, and also Congcong Wen for paper revision.

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

## A    MOBILE CONSTRUCTION TASKS

**Mobile construction robots**.  Recently we have seen a rising trend of 3D printing using mobile robots all around the globe for construction and manufacturing (Werfel et al., 2014; Jokic et al., 2014; Ardiny et al., 2015; Nan, 2015; Marques et al., 2017; Buchli et al., 2018; Zhang et al., 2018; Melenbrink et al., 2020).  All of those are carefully engineered systems that either assume some global localization ability or only work for specific scenarios, which restricts their feasibility in large scale. Moreover, none of them address the aforementioned challenge from a theoretical perspective. To stimulate both robot localization and planning and deep RL research community to work on this problem, we design an efficient mobile construction simulation environment with a series of tasks in 1/2/3D grid worlds.  Several types of goal design $D$ used in each task of 1/2/3D environments are shown in Figure A.2.  We also show some real world examples of mobile construction tasks in Figure A.1.

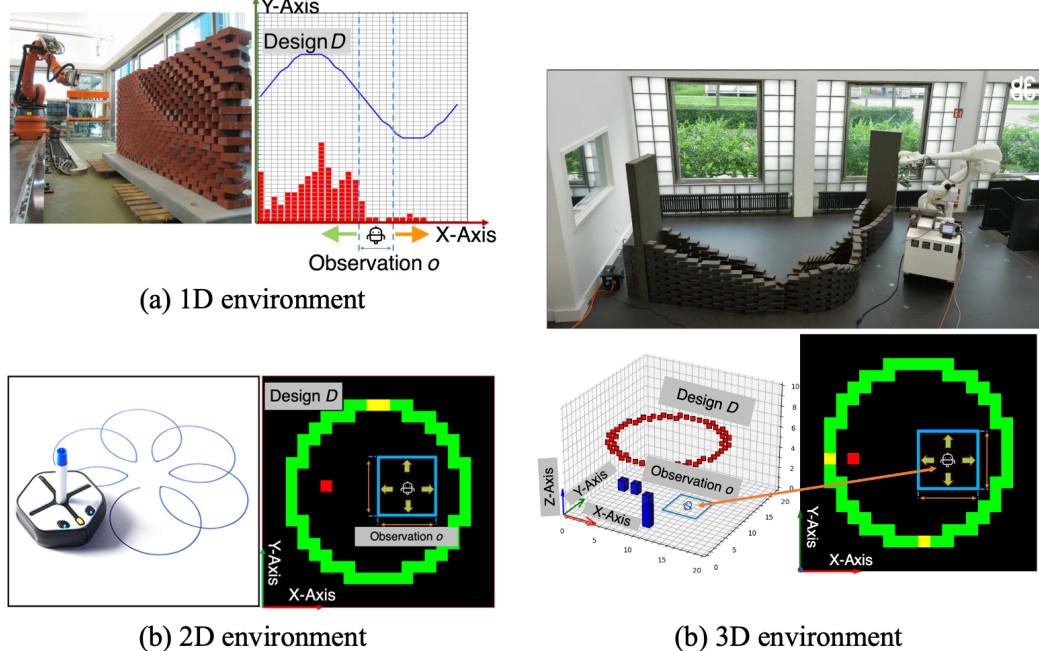

(a) 1D environment

(b) 2D environment                     (b) 3D environment

Figure A.1: a) A real world example of 1D environment (left), a programmed wall project (Bonswetch et al., 2006), and 1D grid world environment (right). b) A real world example of 2D environment iRobot® (left), a painting robot and 2D grid world environment (right). c) A real world example of 3D environment, mobile robot construction project (Giftthaler et al., 2017) (top) and 3D grid world environment (bottom).

For the 1D constant tasks, we consider three types of shape designs $D \in \mathbb{R}^W$: sin function curve, Gaussian curve and Step function curve (see Figure A.2a to Figure A.2c).  For the dynamic tasks, we generate $D$ (Figure A.2d) based on the following equation: $D = a\sin(bx + c)$, where $a \sim U(3, 12), b \in \{1, 2, 3\}$, and $c \sim U(-\pi, \pi)$. The coefficients $a, b$ and $c$ are chosen randomly for each episode. For 2D variable tasks shown in Figures A.2g and A.2h, three vertexes of a triangle are randomly picked within the grid world.  The setup of designs $D$ in 3D is similar to the ones in 2D as shown in Figures A.2i-A.2l.

## B    BASELINE SETUP

Several common model-free and model-based RL baselines, and one handcrafted policy are considered in our paper. Here, we fix the environment constants as follows: (1) for 1D environments, half window size $W_s = 2$, and the environment width $W = 30$; (2) for 2D and 3D environments $W_s = 3$ and $W = H = 20$. Detailed architecture designs and hyperparameter setups for each baseline are explained as follows.

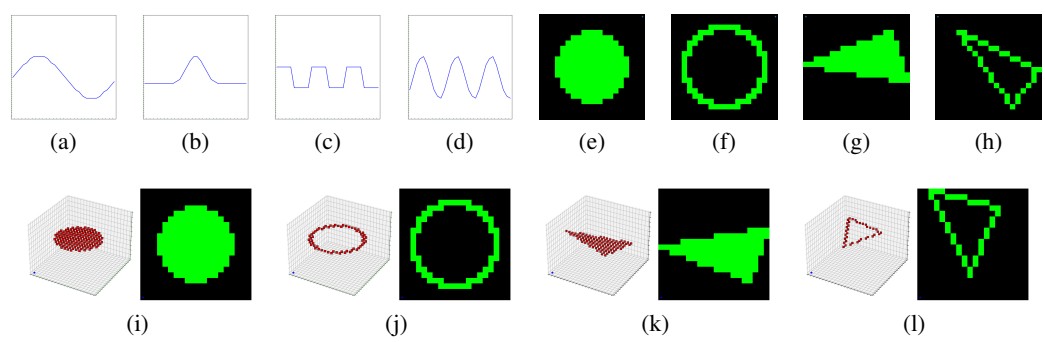

Figure A.2: Example plans. 1) (a-d) shows the curve shapes for 1D tasks. 2) (e/f) are design shapes for 2D constant dense and sparse tasks and (g/h) are 2D variable tasks; 3) (i/j) are 3D constant tasks and (k/l) are 3D variable tasks. The 3D red patches are the top-most surfaces of ground-truth plans.

**DQN**. For the constant design tasks, we use an MLP with three hidden layers containing [64,128,128] nodes with ReLU activation function for each layer. Three convolutional layers with kernel size of 3, and ReLU activation functions are used to convert the ground-truth $D$ to feature vector for variable design tasks. We train DQN on each task for 3,000 episodes. Batch size is 2,000, and replay buffer size 50,000. For tuning the learning rate, we try a relatively small value 1e-8 to make sure convergence initially and gradually increase it until finding a proper value. Additionally, instead of only using the current frame, we try to stack ten frames of historical observations in the replay buffer. This is similar to how the original DQN handles history information for Atari games (Mnih et al., 2013), but no significant difference was found.

**DRQN**. We benchmark DRQN by simply adding one recurrent LSTM layer to the Q network in the DQN. The hidden state dimension is 256 for LSTM layer for all tasks. We train it for 10,000 episodes with batch size of 64 and replay memory size of 1,000.

**DRQN+Hindsight**. We augment the DRQN baseline with hindsight experience replay (Andrychowicz et al., 2017) as another baseline. At the end of each training episode, the transitions $\langle o_{env}^t, a^t, \mathcal{R}(s^t, a^t; D), o_{env}^{t+1}\rangle$ of each time step $t$ will be relabeled as $\langle o_{env}^t, a^t, \mathcal{R}(s^t, a^t; G^T), o_{env}^{t+1}\rangle$, where we change the $D$ to the grid state $G$ at the terminate step $T$. Both transitions are stored into the replay buffer. We train this DRQN+Hindsight for 10,000 episodes with batch size of 64 and replay memory size of 1,000.

**PPO**. We benchmark PPO in discrete settings using the Stable Baselines implementation (Hill et al., 2018). We train PPO for 10 million time steps with a shared network of 3 layers of 512 neurons with tanh activation function. For the hyperparameters, we use the 1D constant environment to tune the learning rate, the batch size, the number of minibatches size, and the clipping threshold. We found that the most sensitive parameters were the batch size and the minibatch size and chose the following values: $1 \times 10^5$ for the batch size, $1 \times 10^2$ for the number of minibatches, $2.5 \times 10^{-4}$ for the learning rate and $0.1$ for the clipping threshold.

**Rainbow**. For the Rainbow implementation, we use 3 noisy hidden layers (Fortunato et al., 2017) with 128 nodes in each layer, and ReLU nonlinear activation functions. Rainbow has a large set of

| | | 1D | | 2D | | | | 3D | | | |
| | | Constant | Variable | Constant | | Variable | | Constant | | Variable | |
| | | | | Dense | Sparse | Dense | Sparse | Dense | Sparse | Dense | Sparse |
| | Avg | 0.0078 | 0.0123 | 0.0069 | 0.0033 | 0.0040 | 0.0017 | 0.0064 | 0.0072 | 0.006 | 0.0058 |
| Simulation time (seconds) | Max | 0.0136 | 0.023 | 0.0072 | 0.0046 | 0.0065 | 0.0029 | 0.0313 | 0.0316 | 0.0209 | 0.0268 |
| | Stddev | 0.0003 | 0.0006 | 0.00004 | 0.0003 | 0.0011 | 0.0003 | 0.0052 | 0.0051 | 0.0046 | 0.0044 |

Table 5: Simulation time of our environments. We test each simulation environment for 500 episodes of games on Intel(R) Core(TM) i9-9920X CPU @ 3.50GHz using a single thread and report the average, maximum, and standard deviation of the simulation time of each game. Our environments can be simulated faster by batch processing.

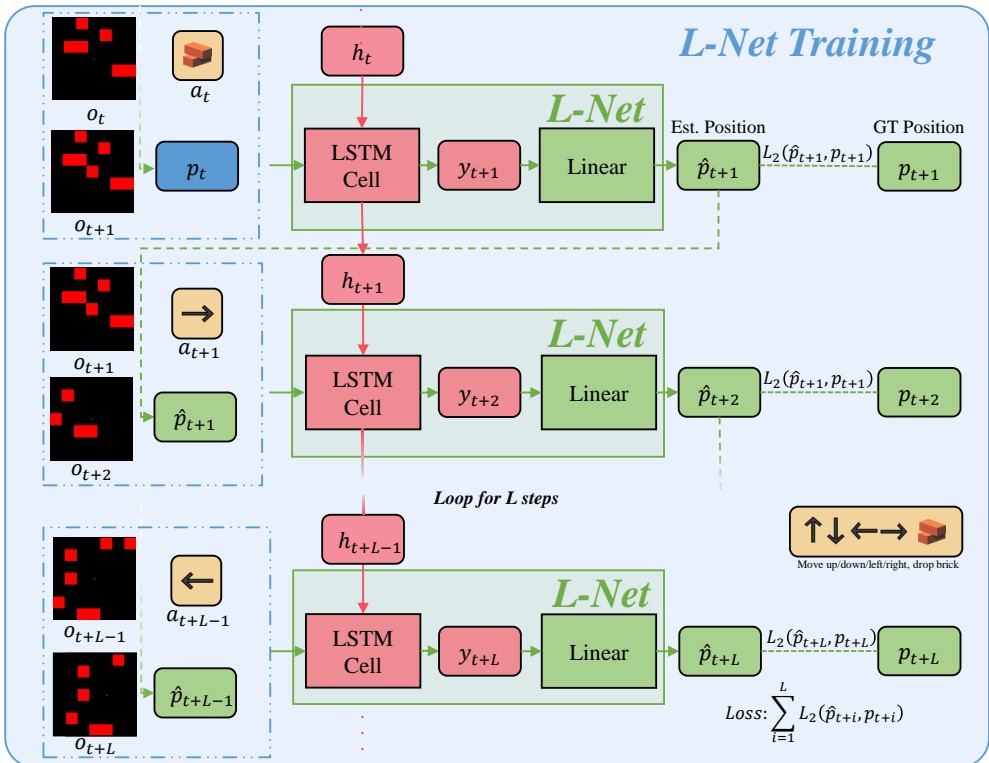

Figure B.1: Illustration of training process of L-Net.

hyperparameters, as each of the six components adds additional hyperparameters. We used those suggested in (Hessel et al., 2017) as a starting point, but they led to poor results on our specific task and environment designs. As a grid search over such a large hyperparameter space was impractical, we used a random search approach. Based on empirical results, the algorithm was most sensitive to learning rate, as well as $V_{min}$, $V_{max}$, and $n_{atoms}$, which define the value distribution support predicted by the distributional Q-network. Generally, a $V_{min}$ value of $-5$, $V_{max}$ of 35, and $n_{atoms} = 101$ provided stable performance, and were chosen heuristically based on an approximate range of discounted rewards possible in our environments. We used a prioritized experience replay buffer of size $1 \times 10^4$, with priority exponent $\omega$ of 0.5, and a starting importance sampling exponent $\beta$ of 0.4. Additionally, we used multi-step returns with $n = 3$, and noisy network $\sigma_0 = 0.1$. Finally, we used a learning rate of $5 \times 10^{-5}$ for 1D and 2D, and $1 \times 10^{-4}$ for 3D environments.

**SAC**. We use the implementation of (Christodoulou, 2019) for SAC in discrete settings which has automatic tuning mechanism for entropy hyperparameters. We use a learning rate $3 \times 10^{-4}$ for target networks for most plans. We use ReLU activation functions for the hidden layers and Softmax for the final layer of the policy network. We use interpolation factor $\tau = 5 \times 10^{-3}$ for target networks and the start steps before running the real policy is 400 with mini batch size 64. We first search the main hyperparameters based on 1D constant case. Next, we use different network architectures for relatively complex 2D and 3D cases with the same hyperparameters as 1D constant case. For 1D environment, we use 2 hidden layers with 64 nodes each for both actor and critic networks. For 2D environment, we use 3 hidden layers with 512 nodes each for the variable design and 3 hidden layers containing [64, 128, 64] nodes for the constant design. A 5-layer network architecture containing [64,128,256,128,64] nodes is applied in the 3D cases.

**DQN+MCTS**. For this model-based method, we simply combine DQN with standard MCTS algorithm, similar to Bapst et al. (2019). In each time step, the tree search method uses the current state $s^t$ as the root node and the latest Q network as a guide to evaluate values for each node of the search tree. When the search process ends, the action $a^t$ is selected based on the highest visit counts among

all possible actions from the root node. In all experiments, we use the true environment simulator as our transition model. We use same architecture design for each task as we use in the DQN experiments. The rollout of each tree search is 20 and the Upper Confidence Bound (UCB) (Auer et al., 2002) is used to balance the exploration and exploitation in the search process. The UCB constant is set to 0.5 for each experiment. We use learning rate $1 \times 10^{-3}$, batch size 2000, and replay buffer size 50,000 for all tasks and train the DQN on each task for 3000 episodes.

**Handcrafted policy.** Besides these RL baselines, we also consider a handcrafted policy with basic localization and planning modules. The localization module borrows the idea of SLAM which uses the common features between successive frames to localize the robot. For the planning modules, we simply let the robot move to the nearest empty grid which should be built. The detail handcrafted algorithm is shown in Algorithm 1. The algorithm is divided into two sub-modules, localization and planning. Localization receives current location $l_t$, current observation $o_t$ and next observation $o_{t+1}$ to determine next position $l_{t+1}$ by finding the common features from two successive observations $o_t$ and $o_{t+1}$. We borrow this idea from visual SLAM which uses similar mechanism to localize. The planning function will find all candidate target location $\mathcal{L}_{\text{candidate}}$ for next step by comparing the current observation with the ground truth design $D$. Based on these candidate positions, the agent will decide whether to move to nearest candidate position or build at current position. The priority action space $A_{\text{prior}}$ is used to decide the action $a$ when the robot does not receive any specific action commend from the nearest planning policy. This priority action will change when robot touches the boundary of the environment. This mechanism helps robot explore more spaces of the unknown environment.

**Human**. For assessing human performance, we made a simple GUI game (details and video samples of the game in supplementary material) that allows a human player to attempt our tasks in identical environments with the same limitations of partial-observability and step size uncertainty. They act as the agent, using the AR-ROW keys to maneuver, and SPACE to drop a brick. The human baseline GUI game is shown in Figure B.2. In constant task environments, human players are required to complete the task without the access to the ground-truth design, while in variable task environments, they can reference the current ground truth design. Additionally, players can toggle between training or evaluation mode. In training mode, they can view per-step reward as well as their cumulative

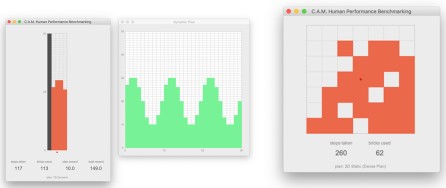

(a) Training (1D).  (b) Evaluation (2D).

Figure B.2: Game GUI for measuring human performance. (a) 1D Variable design: user can see the ground truth design in dynamic environments. (b) 2D Constant design in *evaluation mode*: only step and brick count are shown, while rewards are hidden.

reward over the episode, whereas in evaluation mode, they can only see the number of bricks used and steps taken. For each episode played, players reported their episode-IoU.

## C    BENCHMARK RESULT

In this section, we show the quantitative results of each baseline on all tasks in Table 7. We also explain imitation learning and the Rainbow method ablation study here.

**Learning from expert experience.** Because most of the model-free and model-based RL methods cannot perform well on 2D and 3D tasks due to the aforementioned challenges, we are curious to study whether learning from expert experience could be one potential solution to mobile construction tasks. Therefore, we collect expert experiences on 2D constant sparse and variable dense tasks *via our handcrafted policy* which performs better than the RL methods. Then we train the DRQN baseline with these (instead of random) experiences in the initial replay buffer. Table 6 shows that

|  | IoU | 2D | |
|---|---|---|---|
|  |  | Constant sparse | Variable dense |
| DRQN | Avg | 0.538 | 0.016 |
|  | Stddev | 0.078 | 0.014 |
|  | Min | 0.2 | 0 |
| +Expert experience(↑) | Avg | 0.633 | 0.3 |
|  | Stddev | 0.054 | 0.08 |
|  | Min | 0.303 | 0.12 |

Table 6: Quantitative results of training DRQN with expert experience.

learning from expert can indeed boost the performance of DRQN, especially for the worst performing cases.

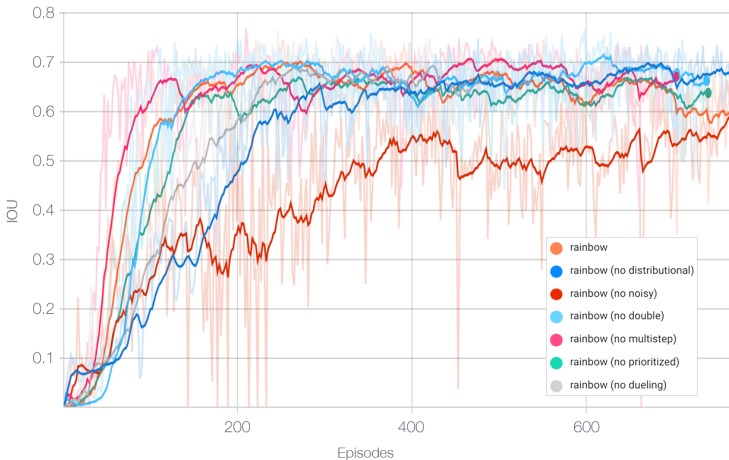

Figure C.1: Training history of IoU: the full Rainbow algorithm does not always outperform its pruned configurations.

**Rainbow method ablation study.** We observed similar performance across most tasks when comparing the base DQN algorithm with DQN plus Rainbow. To better understand the effect of each individual extension on the whole, we conducted six separate runs: in each, we removed one extension from the complete Rainbow algorithm, similar to (Hessel et al., 2017). Figure C.1 displays the IoU running average over training in the 2D variable environment for each of these pruned configurations. Overall, we find that the complete Rainbow algorithm initially learns *faster* than all but one of the pruned configurations, suggesting that the combination of optimizations indeed leads to better sample-efficiency in the learning process. Removing noisy networks (reverting to an epsilon-greedy exploration approach) led to the largest decrease in learning efficiency. Over a longer horizon, the top testing performance plateaus to nearly the same across all configurations. In the case of distributional learning, multi-step learning, and double DQN, removing each individual actually improved top test performance, from 0.676 to 0.685, 0.701, and 0.714 respectively. While this study is not exhaustive, it suggests that the Rainbow algorithm does not easily generalize from the Arcade Learning Environment (Bellemare et al., 2013) (which it was designed for) to our specific tasks.

**Construction accuracy with respect to position estimation error.** We also conduct an experiment to test how the position prediction precision of L-Net could effect the construction performance of DRQN agent. We test DRQN w/ L-Net on 2D variable sparse task for 500 times and get average L2 distance between predicted and GT positions and IoU of each game. As shown in the Figure C.2, we find that the construction performance of the policy is expected to be effected by the accuracy of position estimation by L-Net. With higher localization error, the construction will also be poorer. This indicates that designing an appropriate localization module is key path to solving mobile construction tasks.

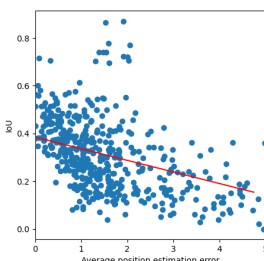

Figure C.2: Results of construction accuracy with respect to position estimation error. The red line shows linear relation between two examined values

| | IoU | 1D | | 2D | | | | 3D | | | |
|---|---|---|---|---|---|---|---|---|---|---|---|
| | | Constant | Variable | Constant | | Variable | | Constant | | Variable | |
| | | | | Dense | Sparse | Dense | Sparse | Dense | Sparse | Dense | Sparse |
| **Human** | Avg | 0.83 | 0.881 | 0.803 | 0.606 | **0.853** | **0.93** | **0.843** | **0.759** | **0.871** | **0.863** |
| | Min | 0.448 | 0.686 | 0.306 | 0.053 | 0.254 | 0.156 | 0.3 | 0.232 | 0.313 | 0.101 |
| Handcraft | Avg | **0.948** | **0.995** | **0.953** | **0.655** | 0.633 | 0.42 | 0.336 | 0.679 | 0.353 | 0.379 |
| | Min | 0.839 | 0.970 | 0.816 | 0.143 | 0.279 | 0.029 | 0.074 | 0.031 | 0 | 0 |
| | Stddev | 0.03 | 0.013 | 0.034 | 0.177 | 0.218 | 0.249 | 0.09 | 0.221 | 0.178 | 0.267 |
| DQN | Avg | 0.799 | 0.781 | 0.744 | 0.234 | 0.061 | 0.044 | 0.024 | 0.022 | | |
| | Min | 0.439 | 0.353 | 0.164 | 0 | 0 | 0 | 0 | 0 | | |
| | Stddev | 0.061 | 0.092 | 0.075 | 0.052 | 0.060 | 0.042 | 0.014 | 0.013 | | |
| DRQN | Avg | 0.861 | 0.850 | 0.818 | 0.538 | 0.427 | 0.068 | 0.071 | 0.009 | | |
| | Min | 0.67 | 0.502 | 0.724 | 0.2 | 0 | 0 | 0 | 0 | | |
| | Stddev | 0.034 | 0.075 | 0.032 | 0.078 | 0.165 | 0.053 | 0.048 | 0.015 | | |
| DRQN+Hindsight | Avg | 0.842 | 0.804 | 0.645 | 0.191 | 0.361 | 0.045 | 0.069 | 0.054 | | |
| | Min | 0.67 | 0.467 | 0.545 | 0.083 | 0 | 0 | 0.018 | 0 | | |
| | Stddev | 0.034 | 0.075 | 0.033 | 0.036 | 0.171 | 0.045 | 0.023 | 0.02 | | |
| SAC | Avg | 0.424 | 0.438 | 0.142 | 0.14 | 0.053 | 0.021 | 0.013 | 0.009 | | |
| | Min | 0.189 | 0.136 | 0.06 | 0.023 | 0 | 0 | 0 | 0 | | |
| | Stddev | 0.102 | 0.13 | 0.034 | 0.019 | 0.047 | 0.033 | 0.011 | 0.014 | | |
| PPO | Avg | 0.788 | 0.794 | 0.65 | 0.294 | 0.358 | 0.140 | 0.317 | | 0.007 | |
| | Min | 0.57 | 0.446 | 0.457 | 0.179 | 0 | 0 | 0.04 | | 0 | |
| | Stddev | 0.039 | 0.065 | 0.055 | 0.038 | 0.114 | 0.068 | 0.092 | | 0.012 | |
| Rainbow | Avg | 0.823 | 0.792 | 0.725 | 0.507 | 0.072 | 0.021 | 0.112 | | | |
| | Min | 0.533 | 0.484 | 0.175 | 0.172 | 0 | 0 | 0.022 | | | |
| | Stddev | 0.064 | 0.061 | 0.093 | 0.093 | 0.087 | 0.035 | 0.038 | | | |
| DQN+MCTS | Avg | 0.745 | 0.738 | 0.752 | 0.154 | 0.135 | 0.020 | 0.032 | 0.015 | 0.013 | 0.007 |
| | Min | 0.059 | 0.313 | 0.587 | 0.031 | 0 | 0 | 0.013 | 0 | 0 | 0 |
| | Stddev | 0.109 | 0.084 | 0.075 | 0.046 | 0.106 | 0.030 | 0.015 | 0.01 | 0.02 | 0.012 |
| DQN w/ L-Net | Avg | 0.829 | 0.799 | 0.869 | 0.662 | 0.687 | 0.065 | 0.098 | 0.083 | | |
| | Min | 0.388 | 0.366 | 0.693 | 0.246 | 0.069 | 0 | 0.015 | 0.008 | | |
| | Stddev | 0.043 | 0.072 | 0.038 | 0.094 | 0.132 | 0.066 | 0.053 | 0.024 | | |
| DRQN w/ L-Net | Avg | 0.896 | 0.863 | 0.854 | 0.684 | 0.7 | 0.301 | 0.374 | 0.315 | 0.143 | 0.053 |
| | Min | 0.624 | 0.38 | 0.696 | 0.289 | 0.132 | 0 | 0.072 | 0.049 | 0 | 0 |
| | Stddev | 0.059 | 0.079 | 0.036 | 0.091 | 0.137 | 0.151 | 0.107 | 0.095 | 0.076 | 0.041 |

Table 7: Benchmark quantitative results. Empty cells indicate the agents failed at the these task without learning any control policy. Blue & purple represent best performance of learning-based methods w/ & w/o L-Net.

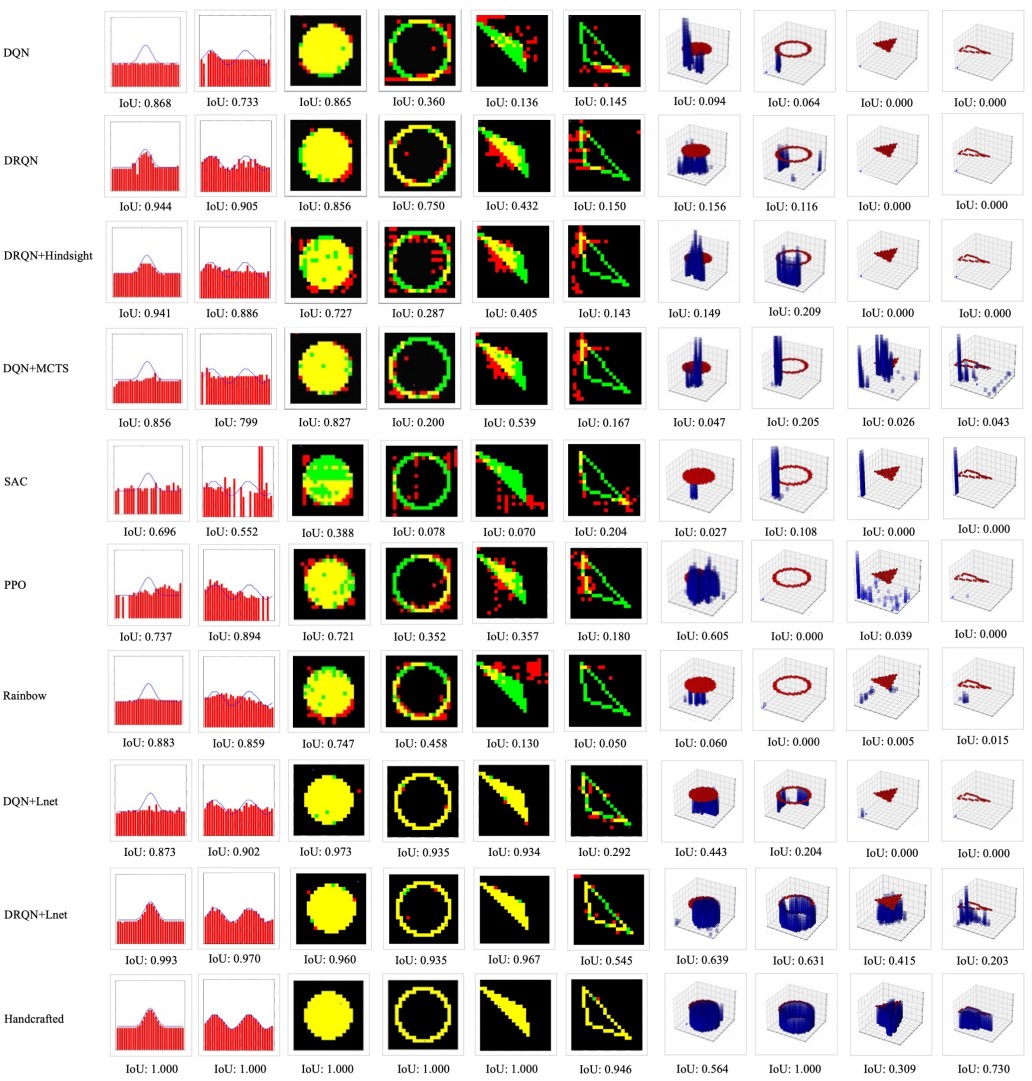

Figure C.3: The *best testing cases* of baselines and our method on all tasks.

---

**Algorithm 1** Handcrafted Policy

---

1: **function** MAIN
2:     Initialize environment and get initial observation $o_{t=0}$
3:     Get initial location $p_{t=0}$
4:     Initialize priority action space $\mathcal{A}_{\text{prior}}$
5:     **for** $t \leftarrow 0, N_{smax}$ **do**
6:         $a_t, \mathcal{A}_{\text{prior}}$=PLANNING$(p_t, o_t, \mathcal{A}_{\text{prior}})$
7:         Execute $a_t$ and obtain next observation $o_{t+1}$, reward, and terminal sign Done from environment.
8:         $p_{t+1}$ = LOCALIZATION$(p_t, o_t, o_{t+1}, a_t)$
9:         **if** Done **then**
10:           Break

11: **function** LOCALIZATION$(p_t,o_t,o_{t+1},a_t)$
12:     Common Feature = FEATUREMATCHING$(o_t, o_{t+1})$
13:     **if** Common Feature is empty **or** $o_t = o_{t+1}$ **then**
14:         Determine $p_{t+1}$ using Odometry, here we assume step size is always 1. **return** $p_{t+1}$
15:     **else**
16:         Determine step size using Common Feature.
17:         Calculate $p_{t+1}$ using step size and $a_t$. **return** $p_{t+1}$

18: **function** PLANNING$(p_t, o_t, \mathcal{A}_{\text{prior}})$
19:     $\mathcal{P}_{\text{candidate}}$ = COMPARE$(o_t, D(p_t))$
20:     Update $\mathcal{A}_{\text{prior}}$ based on the current boundary condition
21:     **if** $\mathcal{P}_{\text{candidate}}$ is empty **then**
22:         $a_t$ is random sampled from $\mathcal{A}_{\text{prior}}$ **return** $a_t, \mathcal{A}_{\text{prior}}$
23:     **else**
24:         Find nearest location $p_{\text{near}}$ from $\mathcal{P}_{\text{candidate}}$
25:     **if** $p_{\text{near}} = p_t$ **then return** $a$=Drop brick, $\mathcal{A}_{\text{prior}}$
26:     **else**
27:         Determine action $a_t$ based on the corresponding direction of $p_{\text{near}}$ to $p_t$ **return** $a_t, \mathcal{A}_{\text{prior}}$

---

