# OpenReview forum: "Learning Simultaneous Navigation and Construction in Grid Worlds "
_ICLR.cc/2023/Conference — ICLR 2023 poster_

### Official Review · Reviewer_cAFg · 2022-10-25

**Confidence:** 3
**Clarity, Quality, Novelty And Reproducibility:** The writing and clarity of the paper …
**Correctness:** 4
**Technical Novelty And Significance:** 2
**Empirical Novelty And Significance:** 2
**Recommendation:** 6

**Strength And Weaknesses:**

## Strengths
* The authors introduce an interesting problem of mobile construction, and develop a grid world setup for experimenting on this task.
* The authors propose a novel approach of localization via an L-Net and action prediction via Deep Recurrent Q Network.
* The authors include extensive experiments: comparisons against human performance, ablations, and comparisons against standard RL baselines (e.g. PPO, DQN) and hand-crafted policies under different setups (e.g. obstacles, no location uncertainty, etc.)

## Weaknesses
* It's not clear to me that the findings in the grid world experiments will transfer easily to more complicated setups, such as mobile construction and mobile manipulation in the real world, in a 3D physics-based simulator, or in minecraft.
* Moreover, most of today's learning-based approaches for learning-based navigation/mobile manipulation (e.g. Savinov et al. semiparametric topological mapping; learning to explore via active neural SLAM) already utilize localization information (and same with non-learning based approaches ie. SLAM), so the finding that decoupling localization from action is helpful for mobile construction is also not very surprising.

## Notes
* Regarding the reward and adding penalties: "we found this led to more frequent actions of moving instead of dropping bricks". It may be useful to add a "time" penalty for each action that doesn't result in a positive reward so that the agent can be more efficient in construction.

**Summary Of The Paper:**

The authors introduce a new problem of mobile construction, where an autonomous agent must construct a grid world according to a design (target grid world state) that is fed as input to the agent. To tackle this problem, the authors de-couple the approach into (1) learning to localize the agent via an L-Net, and (2) zoom in on the design based on the localization info and feed that, along with the observation, as input to a DNN to predict the actions.

**Summary Of The Review:**

My main concern is how the findings and contributions of this paper can be applied more generally. For example, the authors propose the L-Net as a useful decoupling, but this approach is specific to the grid world setup. In more realistic setups, such as iGibson or Habitat, many navigation approaches already predict localization. Also, the experimental findings which show that explicit localization prediction is helpful for the mobile construction task, and that adding obstacles in the grid worlds reduces performance drastically, are somewhat obvious.

Update: Based on the other reviewers' comments and the authors' responses, I've updated my review to be 6.  I agree with the authors' responses that decoupled representations are helpful for this task, and that the findings are not as obvious as I initially thought.

---

> ### Author Response · Authors · 2022-11-11
> **Our findings could be informative for future research in more realistic settings; and existing learning-based navigations are mostly in static instead of evolving environments.**
>
> Thank you for the constructive feedback and suggestions.
>
> 1. “It's not clear to me that the findings in the grid world experiments will transfer easily to more complicated setups”
>
> Thank you for the comment. Our findings in the grid world can be transferred to more complicated setups as follows, which were not widely known to the community before this paper: (1) we now know that applying DRL in a pure end-to-end manner without decoupled representations could not solve the interdependency challenge even in the grid world settings, therefore future researchers facing the similar challenge in more complex settings do not need to try those baselines again and save time; (2) we now know that even if the environment is actively modified by a mobile agent, localization could still be learned by an RNN-based L-Net which could be readily generalized to more photorealistic settings by adopting more powerful backbone networks such as Vision Transformers; (3) we now know that even if such a learned localization may not be perfect, and was learned in an off-policy manner, it could still boost the performance of DRL in tasks facing the similar challenge.
>
> We agree that the grid world setup simplifies the problem, as we pointed out in the 3rd paragraph of the introduction. Yet our simplification is still non-trivial and our findings are still informative, as agreed by the other reviewers: it allows researchers to appreciate the fundamental challenge of the localization-construction interdependency which should be focused on before addressing more complicated setups; it also allows us to look at the challenge in a general and extensible way.
>
> 2. “finding that decoupling localization from action is helpful for mobile construction is also not very surprising”
>
> Thank you for this comment. We would like to point out that the previous methods you mentioned (e.g., Savinov et al.; Chaplot et al.) are all designed and evaluated in **static** environments. Learning to localize in a static environment is not surprising indeed, however **achieving this in an environment that is constantly being changed** has not been shown in those methods, and it is **not very obvious** due to a common concern that dynamically changing environments could degrade localization performance significantly. Moreover, those previous methods are evaluated by navigation success rates and path length, which is **completely different from our evaluation metric** that measures the construction quality by the design-construction IoU. Thus, those previous methods’ findings are not directly comparable to our work, and **do not trivially tell us how to improve mobile construction.** All these reasons are why we believe our findings are significant and non-trivial for the ICLR community, where researchers might be interested in solving our tasks or similar problems.

---

> ### Author Response · Authors · 2022-11-18
> **Do you have other comments or suggestions?**
>
> We would like to know if we have addressed all your concerns.

---

### Official Review · Reviewer_FZE1 · 2022-10-26

**Confidence:** 4
**Correctness:** 4
**Technical Novelty And Significance:** 3
**Empirical Novelty And Significance:** 3
**Recommendation:** 6

**Clarity, Quality, Novelty And Reproducibility:**

The work is original, although similar ideas for explicit learning localization have been used before, i.e. "Integrating Egocentric Localization for More Realistic Point-Goal Navigation Agents, CoRL 2020".

The work is clear and reproducible, assuming the authors release code for the environment, as they say they will.

**Strength And Weaknesses:**

## Strengths

- The proposed task seems challenging for RL but conceptually quite simple. A nice mix.
- The comparisons to existing methods are well done and thorough.
- The proposed method is effective and the way of incorporating positional information is interesting.
- Ablations are well done.

## Weaknesses

No simulation speed numbers are given for the environment itself. Given that this seems to be a visually simplistic environment, I imagine it can be quite fast, but specific numbers would be very useful.

On that note, it also seems like the environment doesn't support batched simulation. I encourage the authors to think about supporting this as such an environment seems idea for this and a faster environment almost always gains more traction.

**Summary Of The Paper:**

This paper proposes a new challenge for AI -- Mobile Construction. In Mobile Construction an agent is initialized in an environment and tasked with building a given structure. The challenge of the task is that it is long horizon and the agent must operate from local observations.

There are 3 versions of the task that are proposed, 1 in 1D, 1 in 2D, and 1 in 3D. This gives an increasing amount of difficulty.

Towards solving this task, the authors propose to explicitly learn localization and incorporate it into the agent by cropping and transforming the target design to be relative to the agents current location.

The proposed method outperforms all the various baselines in the most challenge 3D case.

**Summary Of The Review:**

Overall, my main concern with this paper is lack of evaluation of the environment's simulation performance. This seems like a key part of characterize given that the environment is one that, at least in my understand, should be able to be simulated very quickly.

---

> ### Author Response · Authors · 2022-11-11
> **Simulation speed statistics are added; and our simulation can use batch processing indeed.**
>
> Thank you for the constructive feedback.
>
> 1. “No simulation speed numbers are given for the environment itself.’
>
> Thank you for this constructive suggestion. We test each environment for 500 episodes of games on Intel(R) Core(TM) i9-9920X CPU @ 3.50GHz using the single thread and report the average, maximum, and standard deviation of the simulation time of each game in supplementary Table 5. Note that our environment can simulate even faster by batch processing.
>
> 2. “it also seems like the environment doesn't support batched simulation”
>
> Our environment is built based on the framework of OpenAI gym by mainly using NumPy. Therefore, our environment naturally supports batch simulation by easily using OpenAI built-in multiprocessing tools (https://alexandervandekleut.github.io/gym-wrappers/). We provide a multi-process script in the supplementary material to indicate how to make our simulation environment batch processing in detail. We have uploaded our code in the supplementary material; please feel free to check our implementation in detail.
>
> 3. “The work is original, although similar ideas for explicit learning localization have been used before, i.e. Integrating Egocentric Localization for More Realistic Point-Goal Navigation Agents, CoRL 2020.”
>
> Thank you for your positive and constructive comment. We agree that the ideas of learning-based localization have been used before, but as we discussed in the related work, all those previous works are learning localization in a static environment while our environments are dynamic and evolving. We have added this paper in the related work and updated our discussion.

---

> ### Author Response · Authors · 2022-11-18
> **Do you have other comments or suggestions?**
>
> We would like to know if we have addressed all your concerns.

---

### Official Review · Reviewer_SRjN · 2022-10-26

**Confidence:** 3
**Correctness:** 4
**Technical Novelty And Significance:** 3
**Empirical Novelty And Significance:** 3
**Recommendation:** 8

**Clarity, Quality, Novelty And Reproducibility:**

The formulation of mobile construction as a POMDP appears to be a novel and significant contribution. It is an interesting new class of relevant problems for future works to continue to explore. However, the algorithmic contribution is only marginally novel; it is based on recent work suggesting that the separation of representation learning and reinforcement learning can be an effective practice, and makes use of two existing architectures to achieve these respective tasks. There is not a particularly compelling argument made for why this should be considered a novel contribution that is interesting in its own right, and not just an engineered solution. Similarly, the proposed POMDP formulation of the problem is quite simplistic and quite closely resembles many other POMDPs / grid-world environments from RL, as noted by the paper, so the formulation's novelty is not particularly exciting or insightful. As such, it feels like a small step towards a more complete and realistic formulation of mobile construction (especially the 1D and 2D cases -- the 3D test scenario is a bit better but still seems like a minor step beyond the 2d scenario).

The quality of the experimental results and analysis is a good strength of the paper. Several strong baselines are compared against and the results (and appendix) demonstrate a wide range of different classes of baselines on the new task. The inclusion of a human baseline is also impressive and highly informative, as is the inclusion of handcrafted task algorithms. The ablation study is also detailed and insightful. The entire results section does an excellent job of providing qualitative and quantitative analysis on the performance of different algorithms and on the challenge of the novel task. In terms of further improvement, more discussion of how the problem might change as it takes into account more real-world complexities would be interesting (the analysis of and comparison to baselines, as well as the ablation study, seem largely specific to the test environments explored).

Overall, the clarity of the paper is good. The figures and tables are all excellent at efficiently and effectively conveying various ideas and results. Unfortunately, the legends and text in most of the figures and tables is unreadable unless zoomed in significantly on a screen. Most figures, and table 2, are completely illegible when printed. The font size needs to be increased significantly in all cases, which likely requires enlarging figures. Work also needs to be put into editing the paper; section 4.2 has some grammatical mistakes (e.g. ''train LSTM", "the similar process") as does the Table 2 caption ("2D matrice"), and there other more minor mistakes elsewhere (e.g. "planning" is often misspelled as "planing").

There is no reproducibility statement included in the main text. I think this would be far more useful to the community if the test cases and baselines were implemented in a way to be easily accessed and reproduced.

**Details Of Ethics Concerns:**

Human trials were used but they do not seem to be cause for ethics concerns

**Strength And Weaknesses:**

Strengths
------------
Novelty
- The mobile construction POMDP is interesting and novel
Significance
- RL for mobile construction is highly relevant to a growing subfield in robotics
- The proposed DRQN+L-Net approach significantly outperforms the baselines in the most challenging test cases
- The problem specification is quite general and extensible for future works to build upon and compare against
- The ablation study is very well done and points towards promising future directions
Clarity
- The paper is mostly clear and concise
- Figures are detailed and effective at visualizing the task and results
- Tables 1, 2, and 4 are effectively summarize a great deal of information clearly
Soundness
- The POMDP formulation, choice of evaluation metric (IoU), and other design decisions seem sound

Weaknesses
-----------------
- Marginal novelty algorithmic contribution
- Proposed POMDP is similar enough to existing grid-world type problems that there are not many totally new/unexpected insights
- Feels like a small step towards a more realistic model/simulation of mobile construction
- Font sizes in tables and figures are way too small (unreadable when printed, hard to read on screen without extreme zoom)
- Some grammatical and spelling issues to address
- "So we removed the penalty for better performance" -- better performance on what algorithm(s)? If this is to improve the performance only of the proposed approach, it seems inappropriate to make it part of the experimental setup.

**Summary Of The Paper:**

This paper presents the mobile construction POMDP to formulate the task of a mobile agent placing blocks according to some design, while using observations of its (dynamic) environment to navigate. The paper introduces various instantiations of, and complications to, this task, such as operating in 1/2/3D settings and with/without obstacles. Finally, the paper demonstrates that a deep recurrent Q-network (DRQN) in concert with a localization network (L-net) outperforms baseline approaches and achieves good results on some test cases but leaves room for future work to succeed in others.

**Summary Of The Review:**

This paper serves as a solid foundation for future works to explore more sophisticated approaches to the mobile construction task, as well as more sophisticated formulations of the task itself that capture more real world complexities. The paper does an excellent job of exploring the performance and limitations of existing approaches, and presents a new and somewhat novel approach that outperforms the baselines. The clarity could use some work, and there is some concern about the reproducibility of the experiments. If these issues were addressed, I would feel quite comfortable upgrading my rating.

Update: Based on author responses and revised manuscript, I am upgrading my rating to an 8. Please see comments for details.

---

> ### Author Response · Authors · 2022-11-11
> **Code uploaded and all results are fully reproducible; font size also fixed.**
>
> Thank you for the constructive feedback. Following your suggestions, we have updated **the font size** in figures and tables, and fixed the grammar mistakes in the paper. Regarding your suggestion about the reproducibility statement, we totally agree with that and we have (1) included a reproducibility statement in the updated paper, and (2) submitted all codes and hyperparameters we used for each experiment along with an instruction and readme file about how to use those in the supplementary material to **guarantee all experimental results in the paper are fully reproducible.**
>
> In addition, we provide some discussion regarding your other comments below.
>
> 1. “Marginal novelty algorithmic contribution”
>
> We agree that decoupling representation learning from reinforcement learning is not new, as we acknowledged in our introduction that we are actually inspired by the recent work “Decoupling representation learning from reinforcement learning, PMLR 2021”. So we did not claim algorithmic contribution regarding this point.
>
> But something interesting that our paper reveals is that, even if the agent is proactively changing the environment geometry, the decoupled representation learning could still be done and achieves better performance when integrated with DRL than baselines that couple both localization and construction inside a single network.
>
> 2. “Proposed POMDP is similar enough to existing grid-world type problems”
>
> Our Table 1 explains the difference between our task and existing tasks that include some grid-world ones. The proposed POMDP may look similar to other grid-world type problems, but our task is the first to highlight the fundamental challenge of the tight interdependence of robot localization and long-term planning for environment modification, and our task is the only one that evaluates an agent’s performance by the environment modification quality. The insights that our work might bring are: localization in dynamically evolving environments can be learned by recurrent networks, and imperfect localization learned in an off-policy manner could still improve DRL’s policy learning.
>
> 3. “Feels like a small step towards a more realistic model”, “In terms of further improvement, more discussion of how the problem might change as it takes into account more real-world complexities would be interesting”
>
> As agreed by Reviewer fEQb, starting from a simple yet non-trivial environment can help us focus on solving the core problem of mobile construction tasks. Indeed, a more realistic setup, such as a photorealistic physics-based simulation, is definitely our ultimate goal as we stated in the future work, and it is what we are working on right now. Adding more real-world complexities would bring the following changes to the problem: it would probably bring more challenges to train the L-Net which needs to take first-person-view imagery observations (instead of the 1D or 2D windowed observations) to estimate the agent’s 3D position and orientation (instead of only the position) in a dynamic and evolving 3D environment; it would also bring the need of more lower-level motion control for the agent to deal with the physics-based environment dynamics, which is not present in our current grid-world simulation; and all these changes could make the learning of the construction policy even more difficult.
>
> However, before the more realistic (yet also more costly) simulation environment is ready for research use, our current simple and extremely fast simulation provides a generic and extensible environment for the community to study the fundamental challenge of the localization-construction interdependency. In fact, the research findings made in this simplified environment could still be very informative for future researchers to study similar problems in the more realistic setting, allowing them to make some quick decisions regarding how to design a solution. Therefore, we believe the step we made in this work is an important one that is beneficial for the ICLR community and beyond.
>
> 4. “So we removed the penalty for better performance" -- better performance on what algorithm(s)? If this is to improve the performance only of the proposed approach, it seems inappropriate to make it part of the experimental setup.”
>
> This change in the reward function is for ALL the 2D environments, thus applied to ALL methods, NOT only to the proposed approach. We are sorry for the confusion, and we have updated the corresponding text description accordingly.

---

> ### Author Response · Authors · 2022-11-18
> **Do you have other comments or suggestions?**
>
> We would like to know if we have addressed all your concerns.

---

> > ### Comment · Reviewer_SRjN · 2022-11-18
> > **Response to Authors**
> >
> > Thank you for addressing my concerns. I think that the font sizes in figures 1, 3, and 4 are still a little smaller than they should be, but this is a minor and easily fixed issue. I'm also glad to see the reproducibility statement, as well as the simulation specifics requested by FZE1.
> >
> > I accept the arguments presented to me and my fellow reviewers regarding the need for a first step in this direction to galvanize research efforts and provide a parameterized toy problem to the ICLR community for quick iteration in exploring the localization-construction relationship; thank you for explaining this clearly and convincingly.
> >
> > Regarding point #4, I was specifically concerned about where you observed that the penalty resulted in the development of worse policies; if it was not observed to be an issue for the other algorithms, it seems unfair to remove the penalty only to help your new policy (even if the penalty was removed for all algorithms). But from the revised text it sounds like this was an issue for the baselines as well. In any case, I think you might try to word that a little more clearly (along the lines of "When using the same penalty in the 2D case, we observed X learned really bad policies, so we removed the penalty from the 2d test case." -- if X="only our algorithm", then that seems like a problem, but if X="many/all of the baselines" then it makes sense).
> >
> > Based on the author response and revised text, I am happy to upgrade my score to an 8.

---

### Official Review · Reviewer_fEQb · 2022-10-27

**Confidence:** 4
**Correctness:** 3
**Technical Novelty And Significance:** 3
**Empirical Novelty And Significance:** 3
**Recommendation:** 8

**Clarity, Quality, Novelty And Reproducibility:**

The paper is well written, and conducted experiments are through. The introduced task is novel, and targets an important problem.

**Strength And Weaknesses:**

Strengths:
1. The introduced task targets an important problem, i.e. localization, and challenges current SLAM and sfM methods due to the dynamic status of the environment caused by construction.
2. Authors evaluate a variety of baselines from the literature, and experiments are thorough. The ablation study on the impact of modifying the given input to the agent is interesting, and allows to better understand the source of current agent limitations.
3. The paper is generally well written.

Weaknesses:
1. Compared with other tasks, e.g. in Embodied AI, the environment the agent evolves in is not photorealistic. However, the simple nature of the task can be a good thing to isolate core problems to be solved before to move to environments that are closer to what humans experience in the real world. Moreover, as shown in the paper, the task can not be considered as trivial as most policies do not get strong performance. I would thus tend not to consider lack of realism as a weakness here.
2. Handcrafted methods beat all other baselines in most cases. However, authors mention they have access to more prior knowledge. I would like authors to elaborate on this, as it is an important concern in this work.

**Summary Of The Paper:**

This paper introduces a new task, mobile construction, to evaluate the abilities of an agent to localise within a dynamic environment. The goal is to navigate an environment to build a structure, either in 1, 2 or 3D specified as a goal. The observation available to the agent is the state of the environment in its current neighbourhood. The fact that agents modify themselves the appearance of the environment lead to challenges in terms of localization and mapping.

Different baselines from the literature are considered, and compared against humans and handcrafted baselines. The main takeaway is that the task is challenging, and thus an interesting direction to pursue. Authors also provide a study of the current limitations of agents, by varying the available given inputs.

**Summary Of The Review:**

This work introduces an interesting task, along with an evaluation of diverse baselines from the literature. Compared with other robotics, Embodied AI tasks that also necessitate localisation and mapping, the observations to the agent are much further from real-world observations. However, this should not be considered as a weakness as it allows to properly isolate the studied capacity, i.e. localization in a dynamic environment. Moreover, the ability of the agent to modify its environment makes it an interesting and challenging task for current SLAM methods.

I am however concerned with the much stronger performance of handcrafted baselines, compared with all other agents, and in particular the agent equipped with the L-Net module introduced in this work. Authors should elaborate on what additional prior knowledge these handcrafted baselines have access to, as it looks to me like the design of L-Net is also guided by some prior knowledge about the task at hand.

---

> ### Author Response · Authors · 2022-11-11
> **The handcrafted baseline uses privileged information while others do not.**
>
> Thank you for the constructive feedback.
>
> 1. “The environment agent evolves in is not photorealistic. However, the simple nature of the task can be a good thing to isolate core problems to be solved before to move to more realistic environments”
>
> We highly appreciate your agreement that starting from a simple yet non-trivial environment can help us focus on solving the core problem of mobile construction tasks. Indeed, a more realistic setup is definitely our ultimate goal as we stated in the future work, and it is what we are working on right now. Before the more photorealistic (yet also more costly) simulation environment is ready for research use, this simple and extremely fast simulation provides a generic and extensible environment for the community to study the fundamental challenge of the localization-construction interdependency, and research findings made in this simplified environment could still be very informative for future researchers to study similar problems in a more realistic setting.
>
> 2. “Needs to elaborate how the handcrafted baseline has access to more prior knowledge. As it looks to me like the design of L-Net is also guided by some prior knowledge about the task at hand.”
>
> **The short answer**: the handcrafted baseline leverages the distribution parameters of the probabilistic transition model 𝓣(p'|p,G,a) that should be unknown, while L-Net and other baselines do NOT know this privileged information. Thus it is expected that the handcrafted baseline would have better performance, although such a solution would NOT be very useful to help us moving into more realistic settings when such information is inaccessible.
>
> **The long answer**: Let’s look at the 1D environment. As we stated in the last paragraph of section 3.1, we simply model the motion uncertainty by sampling the moving distance of one moving action from a uniform distribution over three possible distance values, [1,2,3]. Since the half observation window size is 2, it is easy for the agent to figure out whether its moving distance is 1 or 2 grids by comparing the common portion of the current and the previous observation, or 3 if there is no common observation. Therefore, the better performance of the handcrafted policy relies on the knowledge of all possible distance values and solves localization by enumeration. All the other benchmarked methods including the L-Net do not have access to this privileged information. In reality, this information is much more complicated and not easily accessible and should not be assumed to be known. Also in reality the localization by enumeration would not work because distances are not integers anymore.

---

> > ### Comment · Reviewer_fEQb · 2022-11-15
> > **Response to authors**
> >
> > I thank the authors for addressing my concerns.
> >
> > 1. I agree task simplicity should not be considered as a drawback as it allows to isolate important problems to solve in the first step (this is mainly true when considering agents trained with RL).
> > 2. Thank you for clarifying the nature of the prior knowledge given to the handcrafted baseline. Access to the probabilistic transition model is indeed a strong advantage, and thus explains the difference in performance with other baselines. I would like authors to include their “short answer” and “long answer” in the paper.
> >
> > Answers from authors are relevant, and I have thus updated my recommendation score to 8.

---

### Author Response · Authors · 2022-11-11
**Thank you all for the constructive feedbacks and encouragements.**

We sincerely appreciate all reviewers’ constructive feedback and positive comments: (1) “the formulation of mobile construction as a POMDP appears to be a novel and significant contribution” (Reviewer SRjN); (2) “experiments are thorough and ablation is well done” (Reviewer fEQb and FZE1); (3) “the proposed method is effective and the way of incorporating positional information is interesting.” (Reviewer FZE1) and “proposed approach is novel” (Reviewer cAFg). We are also glad to see that our proposed mobile construction tasks are agreed to be novel and interesting by all reviewers.

Based on the reviewers’ comments, we have revised the manuscript which is highlighted with blue text and submitted the code with the instruction on how to reproduce the experimental results. The major modifications we have made: (1) updating tables and figures with larger font size; (2) adding a reproducibility statement; (3) adding the environment’s simulation speed statistics in supplementary Table 5. We also provide detailed responses to each reviewers’ comments individually.

---

### Decision · Program_Chairs · 2023-01-20

**Decision:**

Accept: poster

**Justification For Why Not Higher Score:**

The paper introduces a new benchmark but no new method.

**Justification For Why Not Lower Score:**

-

**Metareview: Summary, Strengths And Weaknesses:**

This paper introduces a new task, mobile construction in a dynamic environment and tests various baselines against it. The paper has been generally appreciated and a consensus was reached quickly. The reviewers also appreciated well executed evaluations and ablations including qualitative and quantitative analysis on the performance of different algorithms.

On the downside, reviewers noted that the task is not photo-realistic as other navigation tasks are, that generalization to non-grid world environments is not straightforward and that no methodological contribution has been provided.

The AC considered these issues outweighted by the interesting benchmark and the well executed evaluation and recommends acceptance.

**Note From Pc:**

if the above contains the word "oral" or "spotlight" please see: "oral" presentation means -> notable-top-5% and "spotlight" means -> notable-top-25%. As stated in our emails, we are disassociating presentation type from AC recommendations

**Summary Of Ac-Reviewer Meeting:**

No meeting was held